# Subcapital Femoral Neck Fracture in a Professionally Active Patient Undergoing Palliative Treatment for Endothelial Cell-Derived Epithelioid Haemangioendothelioma (EHE)

**DOI:** 10.3390/reports7040111

**Published:** 2024-12-09

**Authors:** Paulina Kluszczyk, Aleksandra Tobiasz, Dawid Szumilas, Mateusz Winder, Jacek Pająk, Robert Kwiatkowski, Jerzy Chudek

**Affiliations:** 1Student Scientific Society, Department of Internal Medicine and Oncological Chemotherapy, Faculty of Medical Sciences in Katowice, Medical University of Silesia, 40-029 Katowice, Poland; tobiasz.aleksandra00@gmail.com; 2Department of Internal Medicine and Oncological Chemotherapy, Faculty of Medical Sciences in Katowice, Medical University of Silesia, 40-029 Katowice, Poland; dawid_szumilas@wp.eu (D.S.); chj@poczta.fm (J.C.); 3Department of Radiology and Nuclear Medicine, Faculty of Medical Sciences in Katowice, Medical University of Silesia, 40-752 Katowice, Poland; mwinder@sum.edu.pl; 4Department of Pathomorphology, Faculty of Medical Sciences in Katowice, Medical University of Silesia in Katowice, 40-027 Katowice, Poland; makpaj@wp.pl; 5Radiotherapy Department, Katowice Oncology Center, 40-074 Katowice, Poland; rkwiat11@wp.pl

**Keywords:** endothelial cell-derived epithelioid haemangioendothelioma, EHE, subcapital fracture, femoral neck fracture

## Abstract

**Background and Clinical Significance:** Femoral neck fracture frequently occurs in the elderly population but may also present in patients diagnosed with primary cancer or bone metastases. A pathological, oligosymptomatic fracture associated with epithelioid haemangioendothelioma (EHE), a rare endothelial cell-derived sarcoma, is uncommon. **Case Presentation:** A 44-year-old patient underwent biopsy procedures three times (2010, 2012, 2013) for a focal lesion of the left ischium, none confirming its malignant nature. The last biopsy revealed a neoplastic tissue with features of discrete dysplasia. The lesion did not undergo medical follow-up for seven consecutive years. In August 2020, the patient presented with right lower limb pain. A CT scan, PET/CT scan, and biopsy confirmed EHE with spindle/sarcomatous features. In November 2020, chemotherapy (5xADIC) started (PET/CT confirmed a partial response). After its completion in July 2021, bone progression occurred and sirolimus-based therapy was started. After 3 months, a small liver metastasis was visualized on PET/CT, which did not result in the termination of treatment. In December 2021, pamidronate-based antiresorptive therapy was started. Liver metastasis remained stable in follow-up CT scans. Due to pelvic and spinal lesions, the patient was assisted by elbow crutches and underwent radiotherapy, remaining professionally active. The patient did not report any trauma, but in August 2023, a subsequent CT scan revealed a subcapital fracture of the left femoral neck in the fusion phase. Due to pelvic changes and the stable nature of the fracture, surgical treatment was abandoned. **Conclusions:** An oligosymptomatic femoral neck fracture, not requiring medical intervention is considered a rare complication of bone cancer.

## 1. Introduction and Clinical Significance

Femoral neck fractures are common among the elderly, primarily due to the synergistic effects of osteoporosis and an increased risk of falls. This condition most often affects women and is typically associated with low-energy trauma [1]. However, the fracture can also occur in patients with bone metastases and primary bone cancers. Tumours destroy the bone by promoting proliferation, hypertrophy, and activity of osteoclasts [2]. Skeletal-related events resulting from bone destruction include bone pain and skeletal fractures [3].

Pathologic fractures of the femoral neck, a weight-bearing portion of the hip joint, cause pain that is exacerbated with ambulation. Most patients present to the orthopaedic surgeon unable to bear weight and walk [4]. These fractures rarely heal on their own since the tumour tissue at the fracture site interferes with bone healing, and most fractures have to be managed with surgical intervention [5].

Epithelioid haemangioendothelioma (EHE) is an extremely rare sarcoma derived from endothelial cells. In the 2002 World Health Organisation (WHO) classification, this cancer was described as a locally aggressive tumour with metastatic potential [6,7,8]. However, since 2013, EHE has been considered a moderate malignancy in the WHO classification [9]. It can be diagnosed at any age, but the average age at diagnosis is usually 20–30 years, based on unique histological, immunohistochemical, and molecular features [9,10]. EHE is four times more common in women than men [6,10]. It exhibits a highly variable clinical presentation, including indolent as well as rapidly progressing high-grade sarcomas. It can be found in various parts of the body (liver, spleen, heart, head and neck, thorax, bones, skin, lungs, and other soft tissues) [11,12], but the most frequent is pulmonary localisation (30%). Bone-limited disease occurs in 14% of the affected patients [10]. In immunohistochemical analysis, the tumour cells are positive for vimentin, endothelial markers (CD31 and CD34, factor VIII-related antigen), and vascular markers such as ERG and FLI1 [13]. Imaging features of the tumour are not characteristic (septated lytic lesions with cortical destruction and periosteal reaction can be observed); nevertheless, X-ray and CT are useful in evaluating bone destruction [14,15]. Surgery is the most effective therapy, however inapplicable for extensive lesions. In patients with extensive involvement, chemotherapy and radiation therapy (RTH) may be used. Due to the rarity and little data on the therapies’ effectiveness, the optimal systemic treatment in EHE remains unestablished [16].

In this article, we present the clinical and radiologic findings of a 44-year-old man with EHE with spindle/sarcomatous features with a low-symptomatic fracture of the left femoral neck.

Clinical Significance: Oligosymptomatic femoral neck fracture belongs to the rare bone complication in epithelioid haemangioendothelioma (EHE) with bone involvement.

## 2. Case Presentation

A 44-year-old professionally active (technician servicing medical radiological equipment) man, managed with sirolimus for EHE, was referred for a routine efficacy assessment CT scan in August 2023. The patient was observed to have a limp and experienced pain when walking for a few years.

Before the diagnosis of EHE, the patient underwent biopsy procedures three times in 2010, 2012, and 2013 due to a focal lesion of the ischium on the left side (Figure 1). During each procedure, the material was collected for histological examination.

Bone cysts were found in the biopsies in 2010 and 2012. Notwithstanding, the next biopsy in 2013 revealed neoplastic tissue with features of discrete dysplasia (consulted at the oncology institute). The lesion was not subject to medical supervision for seven consecutive years.

In August 2020, the patient reported pain in the right lower limb and paraesthesia. A CT scan of the lumbar-sacral spine revealed extensive bone destruction of the left ischium, the iliac plate from the left side of the posterior superior iliac spine, the sacrum with vertebral fractures, and compression of the L1 and L4 nerves. In addition to the previously described lesions, 18F-FDG-PET-CT revealed metabolically active changes in the left humerus, left femur, and right tibia. The lesion on the left humerus was asymptomatic and it was not pursued apart from the PET-CT scan. Based on a CT-guided bone biopsy of the iliac spine, the diagnosis of EHE with spindle/sarcomatous features was established (Figure 2). The fragment of spongy bone presented infiltration with mesenchymal, spindle, and epithelioid cells with high- and intermediate-grade atypia and mitotic figures. Intranuclear and intracytoplasmic vacuoles were shown in several cells. No pathological vascular structures were observed. The lining was loose myxomatous with reticulin (Gordon+) and collagen (Masson+/-) fibres. Immunohistochemical examination revealed atypical tumour cells, with marker details provided in Table 1.

In November 2020, chemotherapy (five cycles ADIC—doxorubicin and dacarbazine infusion-based chemotherapy) was started, with a partial response confirmed by 18F-FDG-PET-CT. During active surveillance in July 2021, bone progression was revealed and sirolimus-based therapy was started according to recommendations from the National Institute of Oncology. After the first 3 months, a small liver metastasis was shown in 18F-FDG-PET-CT, which did not result in the termination of treatment. Supportive therapy with pamidronate 60 mg (Pamifos) infusions was started in December 2021. A stable image of liver metastasis (the target lesion) was observed in repeated follow-up CT scans. Palliative radiation therapy (RTH) was administered due to the pain related to the changes in the pelvis and spine (30 Gy in 10 fractions). The analgesic effect of RTH was mild and the patient was continuously using fentanyl-containing transdermal patches, but without the need to use medications for breakthrough pain. He has remained professionally active as a medical equipment service technician, with the help of elbow crutches. This occupation entails moderate physical activity, imposing a consistent, though not excessive, biomechanical load on the hip joint. This level of demand allowed him to continue working despite his mobility challenges.

Upon presentation in August 2023, a CT scan revealed a subcapital fracture of the left femoral neck in the healing phase (Figure 3). The retrospective reassessment of the previous CT performed in April 2023 demonstrated an early stage of femoral neck fracture at the border of the examination range. There was no history of trauma and only mild exacerbation of pain during weight bearing. After consultation with an orthopaedic surgeon, the treatment was abandoned due to the presence of cancer-related changes in the pelvis, the stable nature of the fracture, and the patient’s compromised mobility, which allowed him to function despite the unnoticed fracture. Furthermore, surgery would likely involve significant challenges, including a questionable prognosis for successful healing. The second course of palliative RTH on the pelvis area (30 Gy in 10 fractions) was administered in April 2024. Currently (Figure 4), the patient is continuing therapy with sirolimus and pamidronate, remaining professionally active (November 2024).

## 3. Discussion

The femoral neck fracture is usually the result of a fall and is accompanied by pain and decreased hip motion. Displaced fractures may present with a shortened and externally rotated lower limb. However, the non-displaced fractures may present without deformity [4]. Radiological diagnosis usually includes an X-ray image of the pelvis and an axial or Launsetin view of the affected hip. A CT scan may be occasionally necessary to diagnose non-displaced fractures [17].

In oncological cases, most often bone cancer is of metastatic origin with metastases commonly coming from the breast, prostate, or lung, and less frequently from the kidney or thyroid [18,19]. The impact of cancer cells is related to the interactions between tumour cell receptors (e.g., CXCR4 and RANKL) and the stromal cells of the bone marrow and bone matrix. These interactions trigger the release of growth factors, cytokines (such as IL-6, IL-8), and angiogenic factors (VEGF), which promote tumour growth, osteoclast activation, and their proliferation, with the subsequent osteolysis [2,18].

Tumour-associated bone destruction may also result from primary bone cancers, the most common of which are giant cell tumours, osteosarcomas, and chondrosarcomas. Although these tumours have different origins, they are characterized by the increased activity of osteoclasts, which mediate bone destruction and may cause bone fractures [20]. During the development of the tumour process within the bone tissue, there is an increase in intraperiosteal pressure at the lesion site, which causes mechanical irritation and stretching of the periosteum that stimulates pain receptors [21,22,23,24].

Osteoporosis may contribute to fractures in bone cancers [19] and bone metastases, especially during treatments with hormonal deprivation therapies e.g., breast and prostate cancers [5,25].

To our knowledge, it is the first description of a femoral neck fracture in a patient with EHE. However, such fractures were described in other sarcomas, also in children [26]. EHE is an ultrarare sarcoma, which can primarily appear in bones. Its incidence is 0.38/10^6^/year [12] and in the French Sarcoma ContiCabase group this pathology accounted for 0.4% of all sarcomas. Nonetheless, EHE is more prevalent in women, and those involving bones are more frequently observed in men, with the highest incidence in the upper limb and spine rather than in the lower limb as presented in our patient [16,27,28].

In the described patient, the metastatic lesion on the left humerus was asymptomatic, and not pursued apart from the PET-CT scan. After starting sirolimus-based therapy, follow-up CT scans were performed to assess metastatic lesions in the liver, considered target points, to monitor the therapy’s effectiveness. Changes in bone metastases during oncological therapy are unreliable for assessment of the effectiveness, except for the appearance of new changes indicating disease progression.

Hip prosthesis is one of the most frequently performed procedures for treating degenerative conditions in the hip joint [29]. Surgical treatment involves stabilization, closed or open reduction with internal fixation, artificial midtarsal plication, and total hip replacement. However, factors such as multiple systemic diseases, specific blood supply to the femoral neck [30], or advanced bone cancer progression increase the difficulty of treating femoral neck fractures [31].

Conservative or surgical treatment should be selected based on the type of femoral neck fracture, bone quality, the patient’s age, comorbidities, and overall health status. Much less frequently undertaken [4], conservative management mainly consists of bed rest and adequate traction [30] and also requires the use of assistive devices such as canes, crutches, or walkers to facilitate ambulation and reduce weight-bearing stress. Additionally, relieving the patient’s pain is the priority [31].

According to the literature, in patients with cancer metastases to long bones, surgical treatment is recommended with the primary goals of reducing pain and facilitating self-care, even in cases of advanced disease. Additionally, rapid restoration of limb function is crucial given the typically limited life expectancy [32]. The mechanical integrity of pathological fracture fixations is considered good and durable in 97% of cases, with complications occurring in 3% of patients [33]. However, contrary to the literature, surgical treatment was abandoned in our patient. This decision was supported by the advanced stage of tumour progression, the high risk of postoperative complications related to the destruction of the hip bone, and the late detection of the fracture already in an advanced stage of healing with limited pain raised by the weight-bearing stress.

EHE may cause bone pain, and rarely pathological fractures attributed to the particular affinity for cortical bone, which may result in significant morbidity [6,10,12,16,34,35]. Nevertheless, our patient did not consult any doctor for acute hip pain (chronic pain existed for years) but instead underwent a routine CT scan, in which a femoral neck fracture was identified in the healing phase. Acute pain was not the first clinical symptom preceding the appearance of radiological changes and the onset of a pathological fracture, as described in the literature [21,22,23,24]. This could have been due to continuous analgesic therapy for cancer-related pain—the patient was diagnosed with EHE three years before the femoral neck fracture was detected.

## 4. Conclusions

A mildly symptomatic fracture of the femoral neck, which does not require emergency medical intervention, is one of the rarely observed bone complications of cancer. Although femoral neck fracture is a relatively common symptom in sarcomas, similar fractures have not yet been reported in the literature as the result of EHE. Moreover, the vast majority of femoral neck fractures require surgical intervention, while the described patient was functionally independent, partially bearing weight on the limb and supporting with elbow crutches despite the lack of surgical treatment.

## Figures and Tables

**Figure 1 reports-07-00111-f001:**
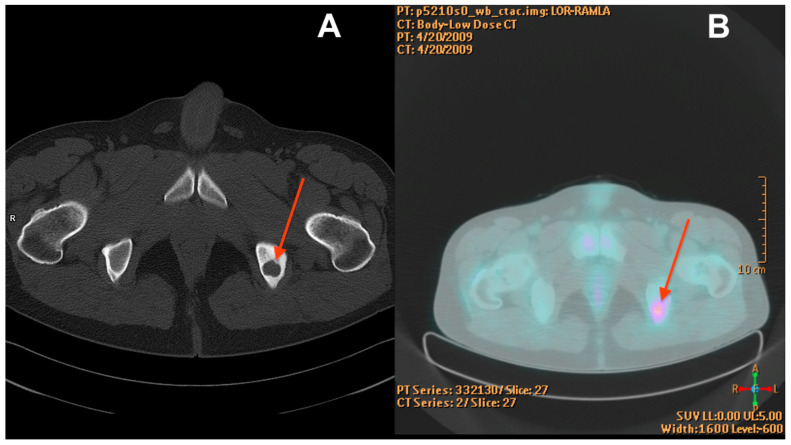
CT scan from 2009 (**A**) showing a well-demarcated osteolytic lesion in the left ischium (red arrow). The mean density of the tumour in the pre-contrast scan was 87 Hounsfield units (HU) and showed moderate enhancement in the subsequent phases reaching 112 HU in the venous phase. PET-CT from 2009 (**B**) with increased radiotracer uptake at the tumour location (red arrow).

**Figure 2 reports-07-00111-f002:**
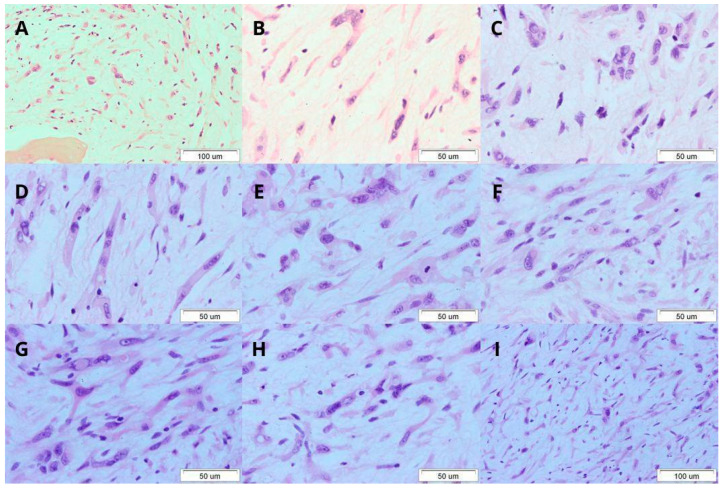
Haematoxylin and eosin (HE) staining (magnification is shown for each picture). (**A**) Intertrabecular space (a bone trabeculae visible in the lower left corner) filled with malignant mesenchymal neoplastic cells with myxomatous lining. Epithelioid, polymorphic tumour cells. (**B**) Epithelioid and spindle-shaped tumour cells. (**C**) Cluster of epithelioid cells. (**D**) Spindle-shaped tumour cells with intracytoplasmic inclusions. (**E**) Tumour cells with intracytoplasmic inclusions. (**F**) Cluster of tumour cells. Erythrocytes are visible in the cytoplasm of one tumour cell (central part of the photo). (**G**) Cluster of tumour cells. In a single cell, intranuclear inclusion and an eosinophil are visible. (**H**) Cluster of tumour cells. A lymphocyte visible in the cytoplasm of a single cell. (**I**) The interbone space filled with malignant mesenchymal neoplasm with myxomatous lining. Polymorphic epithelioid tumour cells.

**Figure 3 reports-07-00111-f003:**
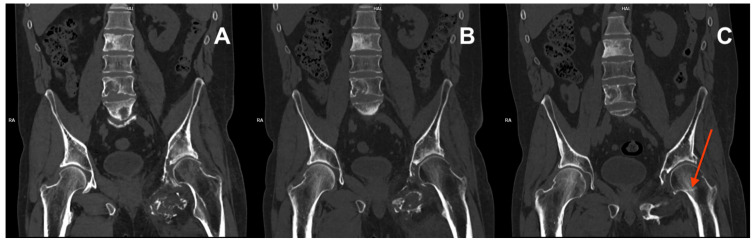
CT scans from September 2022 (**A**), January 2023 (**B**), and April 2023 showing decreased bone density in the left femur and pathological lesions in the vertebrae and left pelvic bones. Early signs of a subcapital fracture of the left femoral neck (red arrow) first seen in the CT from April 2023 (**C**).

**Figure 4 reports-07-00111-f004:**
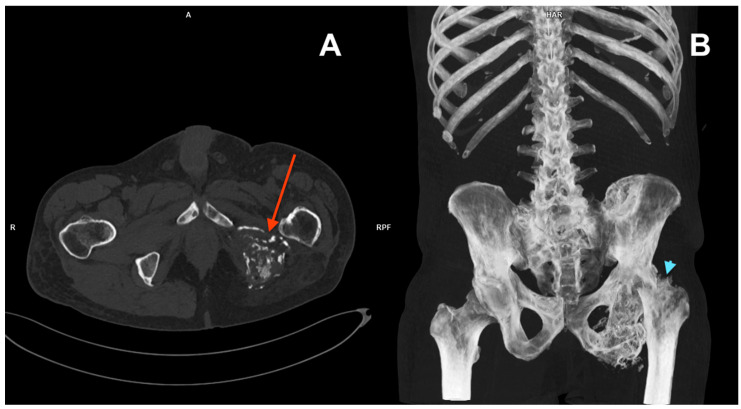
CT scan from August 2023 (**A**) showing extensive lytic infiltration of the left ischium at the initial tumour site (red arrow). Coronal plane maximum intensity projection (MIP) of the same CT (**B**). Impacted fracture of the left femoral neck (blue arrowhead) caused by the neoplastic infiltration. Pathological lesions in the left ischium and hip bone as well as in ribs 10 and 11 on the right side.

**Table 1 reports-07-00111-t001:** Results of immunohistochemical examination of bone biopsy material (2020).

Immunohistochemical Markers	Results in Patient
Vimentin, CD31, CD34, CD99	Positive
B-catenin	Weak
CD163, CD68, CD4, SMA, desmin, CD1a, CD23, CD21, S100, p53, actin, CD117, CD38, CD138, DOG-1, CD99, bcl-2, EMA, Alk-1, CK, CK19	Negative

## Data Availability

Additional patient data can be obtained from the authors upon reasonable request. The data are not publicly available due to privacy concerns.

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
