# Peer review of "Subcapital Femoral Neck Fracture in a Professionally Active Patient Undergoing Palliative Treatment for Endothelial Cell-Derived Epithelioid Haemangioendothelioma (EHE)"

_reports, 2024, doi:10.3390/reports7040111_

Round 1
Reviewer 1 Report
Comments and Suggestions for Authors
Dear Authors
I have a few queries that need to be addressed before considering this submission for publication
1. the previous CT scan showing the progress of the fracture needs to be added
2. the professional activity of the patient is not mentioned to reveal the demand level to the hip joint
3. the fracture is deferred surgery for reasons other than healing state which is the proximity to the lesion area, history of radiotherapy, andso because perse the fracture needs fixation for early and appropriate recovery
4. post radiation, metastatic subcapital fractures are common in the management of any cancers and authors needs to validate their claim of rarity
Author Response
Dear Reviewer,
Thank you for your revision and pointing out the corrections.
1. The previous CT scan showing the progress of the fracture needs to be added
We added the previous CT scans from 2022 and 2023 as Figure 3 showing early signs of the fracture.
2. The professional activity of the patient is not mentioned to reveal the demand level to the hip joint.
Thank you for pointing this out. We apologize for not initially including details on the patient’s professional activity, as this information can be relevant to assessing the demand on the hip joint. The patient is a medical equipment service technician , a profession demanding moderate physical activity. While this profession does require mobility and may add some demand to the hip joint, it does not involve high-impact or physically strenuous activities that would typically exert excessive stress on the joint. We have added this information to the case description (lines 79, 138-142) to provide a more complete context.
3. The fracture is deferred surgery for reasons other than healing state which is the proximity to the lesion area, history of radiotherapy, and so because perse the fracture needs fixation for early and appropriate recovery
We agree that fractures of this nature often benefit from surgical fixation to promote early and effective recovery. However, after consultation with an orthopedic surgeon, surgical intervention was deemed inappropriate for this patient due to several factors. Firstly, cancer-related changes in the pelvis, combined with a history of radiotherapy, would complicate surgical outcomes and healing, making recovery challenging.
Additionally, the patient’s mobility was already compromised due to the bone sarcoma, and he could ambulate even with the fracture, which had gone largely unnoticed by him. Surgical fixation would likely be a difficult and strenuous experience, with a questionable prognosis for successful healing. The fracture was also assessed as stable, further supporting the choice for conservative, non-surgical management.
4. Post radiation, metastatic subcapital fractures are common in the management of any cancers and authors needs to validate their claim of rarity
Primary EHE of the bone is rare, and it represents only 1% of all malignant tumors of the bone. (Kerry G, Marx O, Kraus D, et al. Multifocal epithelioid hemangioendothelioma derived from the spine region: case report and literature review. Case Rep Oncol 2012;5:91-8.
Our case is rare in terms of not the EHE itself, but mainly by a fracture that was not fully symptomatic. We were only able to find one case (which we already quoted in our paper) that shows an oligosymptomatic fracture of the proximal femur (5 - Kong, A. C., Zarate, S. D., & Belzarena, A. C. (2022). Missed pathological femoral neck fracture undergoes spontaneous healing. Radiology Case Reports, 17, 72-76.) therefore for us this validates the rarity of this case.
Kind regards,
Paulina Kluszczyk
Reviewer 2 Report
Comments and Suggestions for Authors
Dear Authors,
I have read a case report regarding a pathological femoral fracture due to EHE. While interesting, several issues need to be addressed.
1) In the introduction, the authors mention that femoral neck is a significant problem for the elderly population. The patient is only 44 years old.
2) What do the authors mean by professionally active patients in the title? There is no significant description in the case presentation regarding "professional" and "level of activity"
3) Line 91 --> on the side of the posterior...., which side?
4) Was the lesion on the left humerus pursued?
5) What is a trepan biopsy?
6) The authors need to explain the cause of the right lower limb pain in this case. Also, why was an MRI not done on this patient?
7) Include the figures of histopathology findings in this patient, especially when there are three rounds of biopsies.
Comments on the Quality of English LanguageThe English language needs extensive changes. Subcephalic is not an acceptable terminology and some sentences feel weird (e.g., The lesion did not undergo medical follow-up for 7 consecutive years and "routine efficacy treatment").
These are only some of the examples and are not extensive. Please consult a professional proofreader.
Author Response
Dear Reviewer,
Thank you for your monumental revision and pointing out the corrections.
1. In the introduction, the authors mention that the femoral neck is a significant problem for the elderly population. The patient is only 44 years old.
We agree that the patient described in our study is not elderly. Our mention that femoral neck fractures are common in elderly patients was intended to emphasize the rarity of this case, as femoral neck fractures in younger individuals are unusual and typically associated with underlying pathological conditions, such as EHE in this instance.
2. What do the authors mean by professionally active patients in the title? There is no significant description in the case presentation regarding "professional" and "level of activity"
Thank you for pointing this out. We apologize for not initially including details on the patient’s professional activity, as this information can be relevant to assessing the demand on the hip joint. The patient is a medical equipment service technician, a profession demanding moderate physical activity. While this profession does require mobility and may add some demand to the hip joint, it does not involve high-impact or physically strenuous activities that would typically exert excessive stress on the joint. We have added this information to the case description (lines 79, 138-142) to provide a more complete context.
3. Line 91 --> on the side of the posterior...., which side?
Thank you for your comment, the change concerned the left side. We have completed the description. (now lines 97-98)
4. Was the lesion on the left humerus pursued?
The lesion on the left humerus was asymptomatic and it was not pursued apart from the PET-CT scan. After starting sirolimus-based therapy follow-up CT scans were performed to assess metastatic lesions in the liver, considered as target points, to monitor the effectiveness of therapy. Changes in bone metastases during oncological therapy are not reliable in assessing the effectiveness of treatment, except for the appearance of new changes indicating disease progression.
5. What is a trepan biopsy?
Thank you for pointing this out. We apologize for the spelling error; "trepan biopsy" was intended to refer to a "trephine biopsy." To enhance clarity, we have revised the text to "bone biopsy." (line 101)
6. The authors need to explain the cause of the right lower limb pain in this case. Also, why was an MRI not done on this patient?
The paresthesia and pain in the patient's right lower limb are likely caused by compression of the L1 and L4 nerves, as observed in the CT scan, and may also be associated with pathological changes in the right tibia, as revealed on PET-CT.
Diagnosis of the right lower limb was performed in another center, where a CT scan was performed. The patient was scheduled to undergo an MRI examination, but due to the diagnostics undertaken at the local hospital and subsequent treatment, the MRI examination was not performed. After starting the therapy, the symptoms of paresthesia disappeared and the pain was well controlled with analgesic treatment.
7. Include the figures of histopathology findings in this patient, especially when there are three rounds of biopsies.
We are sorry but pictures from previous biopsies are not available. We have added pictures from the last biopsy.
The English language needs extensive changes. Subcephalic is not an acceptable terminology and some sentences feel weird (e.g., The lesion did not undergo medical follow-up for 7 consecutive years and "routine efficacy treatment")
Thank you for this indication, we changed terminology like ‘subcephalic’ to ‘subcapital’. and this sentence on: The lesion was not subject to medical supervision for seven consecutive years.
Kind regards,
Paulina Kluszczyk
Reviewer 3 Report
Comments and Suggestions for Authors
this report present an extremely rare case of femoral neck fracture due to EHE.
due to advanced disease, patient was treated non surgically
the case report is well presented and worth publication ++
i have some comments to improve the manuscript
change sub head fracture to sub-capital all over the manuscript
add dose of biophophsonate given
put all IHC results in one table to facilitate reading
add pain scale, functional follow-up after radiotherapy in more details at the end of the manuscript.
figure 2 ; add bone window for CT scan , what do you mean by healing phase? bony union ?
163 #Hip endoprosthesis is one of the most frequently performed procedures for treating degenerative conditions in the hip joint [29]# ?? the reference is wrong here. kindly correct it , also the word endoprosthesis used for tumor prosthesis while for degenerative condition , it's better to use hip prosthesis / total hip arthroplasty
give full name of the abbreviation ADIC and why it was chosen (was the case discussed in tumor board) ?
discuss rational of radiotherapy
discuss rationale of ADIC and then sirolimus (why antiangiogenic treatement was not considered ? )
discuss why 3 biopsies was not accurate ? was these biopsies has been evaluated by an experienced pathologist in MSK tumors ?
add histopathological figure if possible
Author Response
Dear Reviewer,
Thank you for your detailed and monumental revision and pointing out the corrections.
- Change sub head fracture to sub-capital all over the manuscript
We have applied all the corrections as indicated.
2. Add dose of biophophsonate given
Thank you for your suggestion, the description has been updated. The patient received pamidronate 60mg (line 132)
3. Put all IHC results in one table to facilitate reading
We have included the IHC results in Table 1. (line 111)
Table 1. Results of immunohistochemical examination of bone biopsy material (2020).
|
Immunohistochemical markers |
Results in patient |
|
vimentin, CD31, CD34, CD99 |
positive |
|
B-catenin |
weak |
|
CD163, CD68, CD4, SMA, desmin, CD1a, CD23, CD21, S100, p53, actin, CD117, CD38, CD138, DOG-1, CD99, bcl-2, EMA, Alk-1, CK, CK19 |
negative |
4. Add pain scale, functional follow-up after radiotherapy in more details at the end of the manuscript.
‘The analgesic effect of the first RTH was mild and the patient was continuously using fentanyl-containing transdermal patches. He has remained a professionally active technician, with the help of elbow crutches” (line 135-139).
‘The second course of palliative RTH on the pelvis area (30Gy in 10 fractions) was administered in April 2024. Currently (Fig.3), the patient is continuing therapy with sirolimus and pamidronate, remaining professionally active (October 2024).” (line 152-155)
Thank you for your suggestion. Instead of a pain scale, we would like to highlight, that the patient was not required to use medications for breakthrough pain after RTH, remaining on constant treatment with transdermal patches containing fentanyl. Before and after RTH, his performance status was rated at ECOG 1 and throughout the entire period the patient remained professionally active ( line 137).
5. figure 2 ; add bone window for CT scan , what do you mean by healing phase? bony union?
New Figure 3 has been added presenting CT scans from 2022 and 2023 in coronal plane and in bone window. The last CT © presents early signs of a fracture. The healing phase of the impacted fracture from 2024 was indeed presenting signs of bony callus formation and remodelling.
6. 163 #Hip endoprosthesis is one of the most frequently performed procedures for treating degenerative conditions in the hip joint [29]# ?? the reference is wrong here. kindly correct it , also the word endoprosthesis used for tumor prosthesis while for degenerative condition, it's better to use hip prosthesis / total hip arthroplasty
We corrected the reference no. 29 in the references.
7. Give full name of the abbreviation ADIC and why it was chosen (was the case discussed in tumor board)?
ADIC stands for doxorubicin and dacarbazine infusion-based chemotherapy. (line 127-128)
and why it was chosen?
Due to very rare occurrence of EHE, there is no established standard of treatment for this condition. The therapy was chosen after consultation with the National Institute of Oncology.
8. discuss rational of radiotherapy
Radiotherapy was administered as a palliative treatment due to the pain related to the changes in the pelvis and spine. We have supplemented this information in the manuscript ( line 134 and 152).
9. Discuss rationale of ADIC and then sirolimus (why antiangiogenic treatment was not considered ?)
Regarding the second line of treatment, the patient was consulted with specialist in National Institute of Oncology. We have followed their recommendation starting therapy with sirolimus.
10. Discuss why 3 biopsies was not accurate ? was these biopsies has been evaluated by an experienced pathologist in MSK tumors ?
The clinical course of EHE can be diverse, from indolent to highly malignant. In the presented case, the initial course was undoubtedly indolent. Initially, three biopsies were performed and none of them (even though the last one was consulted at the oncology institute) gave a clear diagnosis. Histological examination, apart from the evaluation of the specimens, is also based on the clinical manifestation, but it should be emphasized that in the years 2010-2013 the focal lesion in the ischium did not progress.
11.Add histopathological figure if possible.
Histopathological findings were added as Figure 2.
Kind regards,
Paulina Kluszczyk
Round 2
Reviewer 2 Report
Comments and Suggestions for Authors
Dear Authors,
Thank you for the revision. Please include the explanation in number 4 in your manuscript. Thank you.
Author Response
Dear Reviewer,
Thank you for your valuable comment. We have included the explanation in number 4 in our manuscript (line 100 and 196-201).
Kind regards,
Paulina Kluszczyk